# Prevalence, Antimicrobial Resistance, and Characterization of *Staphylococcus aureus* Isolated from Subclinical Bovine Mastitis in East Coast Malaysia

**DOI:** 10.3390/ani12131680

**Published:** 2022-06-29

**Authors:** Shamsaldeen Ibrahim Saeed, Khairun Anisa Mat Yazid, Hidayatul Athirah Hashimy, Siti Khadijah Dzulkifli, Fatihah Nordin, Nik Azmi Nik Him, Mohd Fikry Fahmi bin Omar, Erkihun Aklilu, Maizan Mohamad, Che Wan Salma Zalati, Nor Fadhilah Kamaruzzaman

**Affiliations:** 1Faculty of Veterinary Medicine, University Malaysia Kelantan, Pengkalan Chepa 16100, Kelantan, Malaysia; anisa932@gmail.com (K.A.M.Y.); hidayatul.athirah.hashimy@gmail.com (H.A.H.); sitikhadijah.dz@gmail.com (S.K.D.); fatihah.nordin95@gmail.com (F.N.); erkihun@umk.edu.my (E.A.); maizan.m@umk.edu.my (M.M.); salma.z@umk.edu.my (C.W.S.Z.); 2Faculty of Veterinary Science, University of Nyala, P.O. Box 155, Nyala 63311, South Darfur State, Sudan; 3Pusat Perkhidmatan Industri Tenusu, Jabatan Perkhidmatan Veterinar, Pasir Puteh 16800, Kelantan, Malaysia; nikazmihim@gmail.com; 4Pusat Perkhidmatan Industri Tenusu, Jalan Chamang, Bentong 28700, Pahang, Malaysia; fikryfahmi@yahoo.com

**Keywords:** *S. aureus*, subclinical mastitis, antimicrobial resistance, intracellular bacteria, biofilms

## Abstract

**Simple Summary:**

Subclinical mastitis remains a burden in the dairy industry due to difficulties in its treatment and the economic losses that are associated with it. This study reported the prevalence and characterization of *Staphylococcus aureus* (*S. aureus*) that was isolated from bovine mastitis in dairy farms in East Coast Peninsular Malaysia. The data from this study showed the presence of subclinical mastitis and antimicrobial resistance (AMR) in *S. aureus* that was isolated from milk samples. In addition, *S. aureus* isolates could invade and survive within the bovine mammary epithelial cell in vitro and form biofilms. This feature provides an additional challenge for mastitis treatment.

**Abstract:**

*S. aureus* is the pathogen that is commonly associated with subclinical mastitis, causing significant economic losses to dairy farms. This infection responds poorly to antimicrobial treatment, which could be due to the development of AMR, biofilm formation, and the intracellular invasion of *S. aureus* into bovine mammary cells leading to treatment failure. Thus, it is important to understand the challenge of this problem. Therefore, the present study aims to determine the prevalence, antimicrobial resistance, and characterization of *S*. *aureus* that was isolated from subclinical bovine mastitis in East Coast Malaysia. A total of 235 milk samples from dairy cows were collected from selected farms in Kelantan and Pahang. The samples were subjected to a somatic cell analysis to identify subclinical mastitis, followed by bacteria isolation and antimicrobial susceptibility testing. The isolated *S. aureus* were further analyzed for their ability to form biofilms and invade the bovine mammary epithelial cells (MAC-T cells) in in vitro infections modeling using a gentamicin protection assay. The overall total of 74/235 (31.4%; 95% CI = 0.31; 0.32) of the milk samples demonstrated >200,000 somatic cells/mL, suggesting the presence of subclinical mastitis in the animals. A total of 39/235 (16.5%; 95% CI = 0.16, 0.17) of the milk samples harbored *S. aureus* which demonstrated resistance towards the following antimicrobials: penicillin (18/39, 46%), ampicillin (17/39, 43.6%), oxacillin (12/39, 31%), tetracycline (10/39, 26%), and erythromycin (7/39, 18%). AMR was recorded for a total of (17/39, 43.6%) of *S. aureus* isolates. All isolates formed biofilms, with (8/30, 27%) strongly biofilm-forming, (18/30, 60%) moderately biofilm-forming, and the remaining (4/30, 13%) of isolates weakly biofilm-forming. Interestingly, the AMR isolates appear to produce weak and moderate biofilm. Moreover, (6/20, 30%) of the *S. aureus* isolates were invasive towards MAC-T cells, as indicated by their ability to evade gentamicin treatment. The study demonstrated the presence of AMR, invasiveness, and biofilm formation in *S. aureus* that was isolated from subclinical mastitis. This characteristic presents additional challenges to existing antimicrobial therapy.

## 1. Introduction

Mastitis is a significant disease affecting the dairy industry, reducing animals’ health and milk quality and subsequently reducing farmers’ income [1,2]. *S. aureus* is the most frequent bacteria to be associated with intramammary infection resulting in a persistent chronic infection [3]. Mastitis is the most common reason for antimicrobial use in dairy cattle, and it can be administered mainly via intramammary and intramuscular routes [4]. However, the treatment has become less effective, with only a 10–30% cure rate [5]. The antimicrobials lose their efficacy in mastitis treatment, due, in part to the development of antimicrobial resistance bacteria (AMR) [6,7,8,9,10,11]. In addition to developing resistance to antimicrobials, *S. aureus* that is isolated from mastitis is also known to form biofilms and is able to invade and survive in host cells. This may present an additional challenge for the antimicrobial treatment of mastitis. The presence of *S. aureus* in the bovine mammary epithelial cells therefore represents a privileged reservoir from which re-infection can occur [12], leading to long-term disease progression and recurrent infections [13].

Thus, it is important to understand the challenge of AMR in *S. aureus* that is associated with mastitis. Therefore, the present study aims to determine the prevalence, antimicrobial resistance, and characterization of *S*. *aureus* that is isolated from subclinical bovine mastitis in East Coast Malaysia.

## 2. Material and Methods

### 2.1. Study Area and Sampling

The study was conducted in five districts (Pasir Mas, Pasir Puteh, and Kota Bharu, Raub, and Bentong) in Kelantan and Pahang, which are located on the East Coast of Peninsular Malaysia (Figure 1). These areas have a tropical climate that is characterized by humidity, high rainfall, and uniform temperatures between 25 °C and 37 °C [14]. Ten small commercial dairy farms were included in this research. A total of 235 milk samples were collected, according to the guidelines by the National Mastitis Council [15]. Briefly, the quarter was washed with tap water and dried, then the teat end was swabbed with cotton that was soaked in 70% ethyl alcohol. To avoid environmental contamination, the first three milk streams were discarded. Following that, approximately 10 mL of milk were collected aseptically into sterile universal bottles. The samples were transported on ice to the Zoonotic laboratory, Faculty of Veterinary Medicine, University Malaysia Kelantan. The samples were processed within 24 h of arrival at the laboratory.

### 2.2. Determining the Somatic Cell Count (SCC)

To detect the subclinical mastitis, an SCC analysis was performed for all milk samples using the Fossomatic TM FC 5000 machine (Foss Analytical A/S, Foss Alle 1, DK-3400, Hillerød, Denmark). For each analysis, 7 mL of milk samples were used. A cutoff value of 200,000 cells/mL was used as a guideline to indicate that milk comes from an animal with subclinical mastitis [16].

### 2.3. Isolation and Identification of S. aureus

The isolation and identification of *S. aureus* from the milk were conducted using microbiological and serological methods. Briefly, 1 mL of milk sample was added into 9 mL of tryptic soy broth (TSB, Oxoid, Hampshire, UK) and incubated at 37 °C for 24 h. Two loopfuls from each broth sample were plated on mannitol salt agar (MSA, Oxoid, Hampshire, UK) and 5% sheep blood agar (Oxoid Ltd., Hampshire, UK) and incubated aerobically at 37 °C for 24 h. The colonies were further examined by Gram-stain biochemical methods and finally confirmed using the latex agglutination test using a Staphytect Plus kit (Oxoid, UK). The confirmed isolates of *S. aureus* were preserved at −80 °C in Luria-Bertani (LB medium) containing 20% glycerol until further use.

### 2.4. Antimicrobial Susceptibility Tests

*S. aureus* antimicrobial susceptibility tests were conducted using a disc diffusion method, according to the method by the Clinical Laboratory Standard Institute (CLSI). Briefly, *S. aureus* were cultivated on nutrient agar and incubated at 37 °C for 18 h. Following that, bacterial colonies were suspended in 5 mL of sterile saline and the suspensions were adjusted to a density that was approximately equal to 1.5 × 10^8^ CFU/mL, according to the McFarland standard. The bacterial suspension was inoculated onto Mueller–Hinton agar (MHA) and tested for susceptibility towards the following antimicrobials disc: ampicillin (AMP; 10 μg); penicillin-G (10 units); tetracycline (TE; 30 μg); chloramphenicol (C; 30 μg); streptomycin (S; 10 μg); gentamicin (CN; 10 μg); sulfamethoxazole/trimethoprim (SXT; 25 μg); ciprofloxacin (5 μg); and oxacillin (1 µg). All antimicrobials’ discs were obtained from Oxoid, England. The discs were dispensed on the surface of the medium with a disc dispenser and the bacterial plate was incubated aerobically at 37 °C for 18 h. The results were recorded and interpreted based on the measurement of the inhibition zone diameter (ZD) using caliper, according to the interpretive standards of CLSI [17]. To determine the level of antimicrobial resistance for each individual bacteria isolate, the MAR index was calculated as the ratio between the number of antibiotics to which the isolate showed resistance to the number of antibiotics to which the isolate had been exposed. A MAR index of >0.2 was indicative of multiple antimicrobial resistance bacteria [18].

### 2.5. Mammary Epithelial Cell Cultures (MAC-T Cells) and Growth Condition

MAC-T cells were obtained from Prof Liam Good, Royal Veterinary College, London. The cells were maintained in freezing medium (90% FBS + 10% DMSO) and kept in −80 °C. for experimental purposes. The cells were maintained in 14 mL of Dulbecco’s modified Eagle’s medium (DMEM, Sigma-Aldrich, Hampshire, UK) that was supplemented with 10% fetal bovine serum (FBS, Sigma-Aldrich, UK) and 5% penicillin-streptomycin (Sigma-Aldrich, Hampshire, UK). The cells were maintained at 37 °C in 5% CO_2_. All cell culture work was performed under an aseptic technique in a biological safety cabinet.

### 2.6. Intracellular Invasions of MAC-T Cells by S. aureus

Intracellular invasions of the MAC-T cells (host cells) by *S. aureus* isolates were established using a gentamicin protection procedure, as described by Kamaruzzaman et al. [19]. Briefly, the MAC-T cells were seeded at 1.2 × 10^5^ cells/well in a 12-well plate and cultured overnight in DMEM with 10% FBS, without antimicrobials. In parallel, 1 mL of overnight *S. aureus* culture was centrifuged at 8000 rpm, and the pellet was washed with PBS to remove the bacterial toxin. A total of 1 mL of 1 × 10^7^ CFU/mL of bacteria that was diluted in cell culture media without antimicrobials was added into the cell culture plate for 3 h to allow bacterial invasion. Following that, gentamicin (200 mg/L) was added for another 3 h to kill the extracellular bacteria. Finally, the gentamicin was removed and the cells were washed with PBS to remove the residual antimicrobials. A total of 1 mL of 0.5% Triton X-100 that was diluted in PBS was added to lyse the host cells. The lysed cells were serially diluted in PBS and plated on nutrient agar for enumeration of the intracellular bacteria. The uninfected cells were subjected to the lysis procedure to confirm their sterility.

### 2.7. Biofilm Formation

Biofilm formation assays were performed with *S. aureus* isolates (n = 30) following the protocol that was described in our previous work [19]. Briefly, the bacteria were grown overnight at 37 °C in 5 mL of tryptic soy broth (TSB; Oxoid) for 18 h. Following 18 h of incubation, the bacteria were diluted in TSB that was supplemented with 20% glucose to reach the final concentration of 10^7^ CFU/mL. Aliquots of bacteria (1 mL, 10^7^ CFU/mL) were placed in a 12-well cell culture plate and incubated at 37 °C for 48 h. After incubation, the non-adherent cells were removed and the produced biofilms on the surface of the wells were fixed at 60 °C for 1 h. The biofilm was stained with 0.1% crystal violet solution and rinsed with PBS. The OD was measured at 550 mm using a spectrophotometer. Each assay was performed in triplicate. *S. aureus* 15 AL was used as positive control.

### 2.8. Statistical Analysis

A statistical analysis was performed using a one-way analysis of variance (ANOVA). The data are presented as means ± standard deviation (SD). The level of significance was accepted as *p* ≤ 0.05. For the graph, error bars represent the standard deviations. * *p* ≤ 0.05; *** *p* ≤ 0.001; **** *p* ≤ 0.000; and NS, not significant. All the experiments were performed at least three times.

## 3. Results

### 3.1. Prevalance of Subclinical Mastitis and S. aureus

To determine the prevalence of subclinical mastitis in the dairy farm, milk samples were analyzed for their somatic cells count (SCC). Of 235 samples, 74 (31.4%; 95% CI = 0.31; 0.32) showed a number of cells of more than 200,000 cells/mL, suggesting that the animal contracted subclinical mastitis when the sampling was carried out. The highest subclinical mastitis was recorded in Pahang state from the Bentong and Raub area with a 45.2% and 25.2% occurrence rate, respectively. Meanwhile, the milk samples that were collected from Pasir Mas, Kota Bharu, and Pasir Putih in Kelantan state had lower rates of subclinical mastitis at 9%, 16.6%, and 27%, respectively. The milk samples were further subjected to a microbiological analysis for *S. aureus* isolation. A total of 39/235 (16.5%; 95% CI = 0.16, 0.17) milk samples were positive for *S. aureus* (Table 1).

### 3.2. Antimicrobial Susceptibility Testing

Kirby–Bauer methods on Mueller–Hinton agar were performed to determine the susceptibility of *S. aureus* to the antimicrobials that were tested. *S. aureus* showed resistance to the following antimicrobials: penicillin (18/39, 46%); ampicillin (17/39, 43.6%); oxacillin (12/39, 31%); and tetracycline (10/39, 26%), and (7/39, 18%) were resistant towards erythromycin. In contrast, all *S. aureus* were sensitive (39/39, 100%) towards chloramphenicol, ciprofloxacin, sulfamethoxazole/trimethoprim, and gentamicin. Table 2 summarized the antimicrobial susceptibility of *S. aureus.*

### 3.3. S. aureus AMR Profile and Multiple Antimicrobial Resistance (MAR) Indexes

The AMR profile and MAR index of *S. aureus* isolates are summarized in Table 3. A total of (17/39, 43.6%) of *S. aureus* showed an antimicrobial resistance profile (resistance to >1 antimicrobials). This profile included 17.9% of isolates that were resistant to five antimicrobials; 7.7% with a resistance to four antimicrobials; 5.1% with a resistance to three antimicrobials; and 12.8% with a resistance to two antimicrobials. The most predominant *S. aureus* AMR profile was AMP, E, P, TE, OXA and AMP, P. The MAR index analysis indicated that 41% of AMR isolates had a MAR index > 0.2, indicating that the isolates were multiple antimicrobial-resistant bacteria.

### 3.4. S. aureus Biofilm Formation and Their Association with AMR Profile

The biofilm formation of *S. aureus* (n = 30) showed that all isolates could produce biofilms. A total of 27% of isolates were substantial, 60% were moderate, and the remaining 13% were weak biofilm producers (Figure 2). Table 4 shows the biofilm formation ability of *S. aureus* and its relationship with antimicrobial resistance. Our study showed that 75% and 61% of weak and moderate biofilm producers were AMR, respectively, compared to 12.5% of strong biofilm producers, suggesting that AMR isolates tend to develop weak and moderate biofilms.

### 3.5. Intracellular Invasion of MAC-T Cell by S. aureus

A gentamicin protection assay was performed to determine the *S. aureus* invasion activities toward the bovine mammary epithelial cells. A total of 20 isolates were selected based on their susceptibility to the leading antimicrobial group and subjected to a gentamicin protection assay. Six isolates were found to be invasive towards MAC-T cells as evidenced by their ability to escape gentamicin treatment by infecting the host cells. Following cell lysis, the bacteria were enumerated. These isolates were labeled F31D, F41A, F41B, F51B, F53D, and PBF1 (Figure 3).

## 4. Discussion

Subclinical mastitis remains a burden in the dairy industry due to difficulties in the treatment and the economic losses that are associated with it. This study reported the prevalence of subclinical mastitis and antimicrobial resistance of *S*. *aureus* that was isolated from bovine subclinical mastitis in East Coast Malaysia. This study also investigated the intracellular invasion and biofilm formation ability of the isolates. Our results showed the high prevalence of AMR in *S. aureus* that was isolated from bovine subclinical mastitis, and the isolates were able to invade bovine mammary epithelial cells in in vitro infection modeling. The isolates were also able to form a biofilm, and the AMR isolates tended to form weak and moderate biofilms.

Subclinical mastitis is inherently difficult to detect because the animal shows no clinical signs, and no abnormalities are visible in the milk. The current diagnostic method relies heavily on the CMT test and somatic cell analysis using a laboratory that is usually available in reference laboratories, which limits the effectiveness of early detection of the disease. Several new methods have been developed to allow the early detection of subclinical mastitis. For example, ELISA and lateral flow have been developed for the early detection of the enzyme myeloperoxidase of milk neutrophils, a biomarker for subclinical mastitis [20,21].

In this study, a high prevalence of subclinical mastitis was found in the milk sample based on somatic cell count overall (31.4%; 95% CI = 0.31; 0.32). The prevalence rate differs from one province to another, and this could be due to the farm management and implementation of a mastitis control strategy, which includes the rapid identification and treatment of mastitis cases, the isolation and culling of infected cows, practicing the routine dry cow therapy, and post-milking teat disinfection. A similar finding was reported in the state of Selangor on the West Coast of Peninsular Malaysia. The prevalence of mastitis ranged from 70 to 90% in each farm [22]. Another study that was conducted in farms in Selangor and Johor showed a prevalence of 81% of subclinical mastitis in the animals [23]. The prevalence of *S. aureus* that was associated with mastitis was also found in other countries, with 4.8% in Brazil, 12.5% in Iran, 25.8% in China, 28.6% in Turkey, and 55% in Malaysia [24,25,26,27].

A systematic review and analysis revealed that the overall estimate of worldwide *S. aureus* resistance from 1969 to 2000 was to penicillin, clindamycin, erythromycin, and gentamicin [6,28]. In our study, a high prevalence of AMR to penicillin, ampicillin, oxacillin, tetracycline, and erythromycin was found, consistent with previous studies from the Malaysian states of Johor and Terengganu [29]. In another study in Penang state, *S. aureus* was found to be resistant to the following antimicrobials: penicillin, ampicillin, trimethoprim, cefoxitin, linezolid, clindamycin, erythromycin, and tetracycline [9], and the same trend was found elsewhere. For example, a study that was conducted in China showed that 61.1% of *S. aureus* isolates had multidrug resistance [24]. In another study that was conducted in Ethiopia, *S. aureus* was highly resistant to penicillin and tetracycline [30].

In this study, *S. aureus* that was isolated from milk was able to invade bovine mammary epithelial cells, as evidenced by their ability to evade treatment with gentamicin. This study was supported by the results of other studies reporting the invasion activities of *S. aureus* in bovine mammary cells [31]. It has been reported that the uptake of *S. aureus* into mammalian cells is mediated by a zipper-type mechanism. The fibronectin-binding proteins of the bacterium can bridge to the à5ss1 integrin on mammalian cells to induce its zipper uptake mechanism into host cells [32]. *S. aureus*-related subclinical mastitis has been associated with a weak udder immune response. In general, no solid systemic response has been detected due to activation of the Wnt/B-catenin cascades, leading to the active suppression of NF-KB signaling [33]. These weak immune system responses following *S. aureus* infection resulted in persistent infections in udders.

In this study, all *S. aureus* that were isolated from milk were capable of forming biofilms. Several studies have found that *S. aureus* causing bovine mastitis can form biofilms, which are a critical factor in treatment failure [34,35,36,37]. Bacteria in biofilm structures are inherently difficult to treat with antimicrobials, compared to single cells, because the structure is less permeable to conventional antimicrobials. In addition, the extracellular DNA on the surface of the biofilm could interact with the small molecules of antibiotics and intercept them, preventing them from reaching the individual cells. In addition, the bacteria could exist within the biofilm structure as small colony variants and have better tolerance to antimicrobials, compared to the wild type [38].

Biofilm formation is considered to be part of the bacterial survival strategy. Exposure of bacterial strains to subinhibitory concentrations of certain antimicrobial agents is thought to provoke and induce biofilm formation. Ranieri et al. discussed the proposed mechanism of biofilm formation that is induced by subinhibitory concentrations of antimicrobials, including beta-lactams and quinolone antibiotics, to a range of bacteria, including *S. aureus* [38].

## 5. Conclusions

In conclusion, this study shows that subclinical mastitis is a problem in dairy cows in East Coast Malaysia. The high prevalence of AMR, invasiveness, and ability of *S. aureus* to form biofilms poses additional challenges to existing therapy. Further improvement of the current treatment is needed to improve animal recovery, increase milk production, and thus increase dairy production in Malaysia.

## Figures and Tables

**Figure 1 animals-12-01680-f001:**
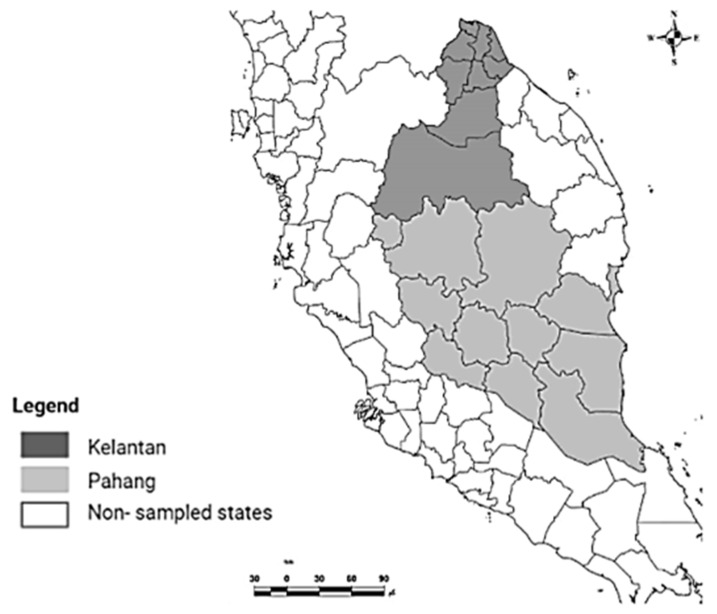
Map of the study area. The samples were collected from dairy farms in Kelantan and Pahang, East Coast of Peninsular Malaysia. The map was created using ArcGIS v. 7 (Esri Inc., Redlands, CA, USA).

**Figure 2 animals-12-01680-f002:**
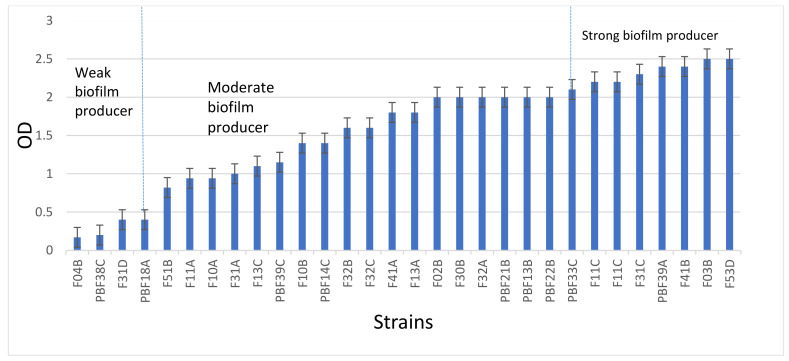
Biofilm formation of *S. aureus* isolates. Based on OD reader the biofilm was classified into 3 categories: strong (OD550 > 2.0), moderate (2.0 > OD550 > 0.4), weak (0.4 > OD550 > 0.1), or negative (OD550 < 0.1). OD 0.1 was set as a cut-off point to distinguish between biofilm producer and non-biofilm producer. Error bars represent standard deviation of triplicates.

**Figure 3 animals-12-01680-f003:**
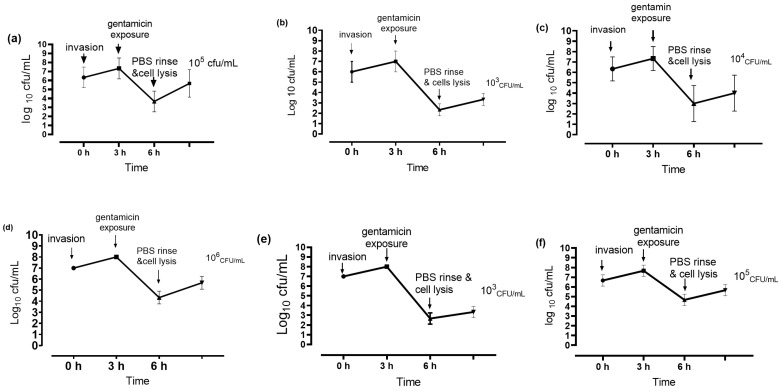
*S. aureus* invasion of MAC-T cells. Each graph (**a**–**f**) shows the survival of invasive isolates of *S. aureus* in MAC-T cells. Lysis of MAC-T cells after gentamicin exposure released approximately 10^6^ CFU/mL of *S. aureus* F41B (d), 10^5^ CFU/mL of F41A (a) and PBF1 (f), 10^4^ CFU/mL of F31D (d), and 10^3^ CFU/mL of F53D (e), and F51B(b). The graphs were generated using GraphPad Prism 8 (San Diego, CA, USA).

**Table 1 animals-12-01680-t001:** The prevalence of subclinical mastitis in different areas in Kelantan and Pahang, Malaysia.

State/locality	No. of samples	Subclinical Mastitis (%)	*S. aureus* isolates (%)
Kelantan			
Pasir Putih	37	10 (27%)	14 (37.8%)
Kota Bharu	12	2 (16.6%)	1(8.3%)
Pasir Mas	11	1 (9.0%)	3 (27%)
Total	60	13 (21.6%)	18 (30%)
Pahang			
Bentong	84	38 (45.2%)	13 (15.5%)
Raub	91	23 (25.2%)	8 (8.7%)
Total	175	61 (34.8)	21 (12%)
Overall	235	74 (31.4%)	39 (16.5%)

**Table 2 animals-12-01680-t002:** Antimicrobial susceptibility of *S. aureus* isolated from bovine subclinical mastitis.

Antimicrobials	Dose (µg)	Breakpoint					
		S (mm)	R (mm)	* n/39	R (%)	I (%)	S (%)
P	10	≥29	≤28	18	46	0	54
AMP	10	≥29	≤28	17	43.6	0	56.4
CN	10	≥23	≤17	0	0	0	100
E	15	≥23	≤13	7	18	0	82
OX	1	≥13	≤10	12	31	0	69
SXT	25	≥16	≤10	0	0	0	100
TE	30	≥23	≤17	10	26	0	74
CIP	5	≥21	≤15	0	0	0	100
C	30	≥18	≤12	0	0	0	100

* n: No. of resistance *S. aureus*, S: susceptible, I: intermediate, R: resistance, P: penicillin, AMP: ampicillin, CN: gentamicin, E: erythromycin, OX: oxacillin, SXT: sulfamethoxazole/trimethoprim, TE: tetracycline, CIP: ciprofloxacin, C: chloramphenicol.

**Table 3 animals-12-01680-t003:** Antimicrobial resistance profile of *S. aureus* isolates from bovine subclinical mastitis.

*S. aureus* Isolates Code	AMR Profile	No. of Antimicrobial Class	MAR Index
F51B	AMP, P	1	0.11
F02B	AMP, P	1	0.11
F30B	AMP, P	1	0.11
F32A	AMP, P	1	0.11
PBF14C	AMP, P	1	0.11
F31A	AMP, P, OXA	1	0.11
PBF38C	AMP, P, OXA	1	0.11
F04B	AMP, P, TE, OXA	2	0.22
PBF39C	AMP, P, TE, OXA	2	0.22
PBF18A	AMP, P, TE, OXA	2	0.22
F31D	AMP, E, P, TE, OXA	3	0.33
F32C	AMP, E, P, TE, OXA	3	0.33
F32B	AMP, E, P, TE, OXA	3	0.33
F41A	AMP, E, P, TE, OXA	3	0.33
F41B	AMP, E, P, TE, OXA	3	0.33
PBF2IB	AMP, E, P, TE, OXA	3	0.33
PBF33C	AMP, E, P, TE, OXA	3	0.33

MAR: multiple antibiotics resistance, P: penicillin, AMP: ampicillin, CN: gentamicin, E: erythromycin, OX: oxacillin, SXT: sulfamethoxazole/trimethoprim, TE: tetracycline, CIP: ciprofloxacin, C: chloramphenicol.

**Table 4 animals-12-01680-t004:** Association between *S. aureus* biofilm formation and antimicrobial susceptibility profile.

NO	*S. aureus* Isolates	Biofilm Production Assay	Biofilm Production Ability	Antimicrobial Susceptibility								
				P	AMP	CN	E	OX	SXT	TE	CIP	C
1.	F04B	0.17	+	R	R	S	S	R	S	R	S	S
2.	PBF38C	0.2	+	R	R	S	S	R	S	S	S	S
3.	F31D	0.4	+	R	R	S	R	R	S	R	S	S
4.	PBF18A	0.4	+	R	R	S	S	R	S	R	S	S
5.	F51B	0.82	++	R	R	S	S	S	S	S	S	S
6.	F11A	0.94	++	S	S	S	S	S	S	S	S	S
7.	F10A	0.94	++	S	S	S	S	S	S	S	S	S
8.	F31A	1.0	++	R	R	S	R	S	S	S	S	S
9.	F13C	1.1	++	S	S	S	S	S	S	S	S	S
10.	PBF39C	1.15	++	R	R	S	S	R	S	R	S	S
11.	F10B	1.4	++	S	S	S	S	S	S	S	S	S
12.	PBF14C	1.4	++	R	R	S	S	S	S	S	S	S
13.	F32B	1.6	++	R	R	S	R	R	S	R	S	S
14.	F32C	1.6	++	R	R	S	R	R	S	R	S	S
15.	F41A	1.8	++	R	R	S	R	R	S	R	S	S
16.	F13A	1.8	++	S	S	S	S	S	S	S	S	S
17.	F02B	2.0	++	R	R	S	S	S	S	S	S	S
18.	F30B	2.0	++	R	R	S	S	S	S	S	S	S
19.	F32A	2.0	++	R	R	S	S	S	S	S	S	S
20.	PBF21B	2.0	++	R	R	S	R	R	S	R	S	S
21.	PBF13B	2.0	++	S	S	S	S	S	S	S	S	S
22.	PBF22B	2.0	++	S	S	S	S	S	S	S	S	S
23.	PBF33C	2.1	+++	R	R	S	R	R	S	R	S	S
24.	F11C	2.2	+++	S	S	S	S	S	S	S	S	S
25.	F11C	2.2	+++	S	S	S	S	S	S	S	S	S
26.	F31C	2.3	+++	S	S	S	S	S	S	S	S	S
27.	PBF39A	2.4	+++	S	S	S	S	S	S	S	S	S
28.	F41B	2.4	+++	R	R	S	R	R	S	R	S	S
29.	F03B	2.5	+++	S	S	S	S	S	S	S	S	S
30.	F53D	2.5	+++	S	S	S	S	S	S	S	S	S

P: penicillin, AMP: ampicillin, CN: gentamicin, E: erythromycin, OX: oxacillin, SXT: sulfamethoxazole/trimethoprim, TE: tetracycline, CIP: ciprofloxacin, C: chloramphenicol, S: sensitive, R: resistance, +++: strong biofilm, ++ moderate biofilm, + weak biofilm.

## Data Availability

All obtained data from this study were included in the manuscript.

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
