# Peer review of "Prevalence, Antimicrobial Resistance, and Characterization of Staphylococcus aureus Isolated from Subclinical Bovine Mastitis in East Coast Malaysia"

_animals, 2022, doi:10.3390/ani12131680_

Round 1

Reviewer 1 Report

This is an interesting study, which adds some nice additional data to the field of microbiology, AMR and mastitis which are rapidly growing fields. The science is robust, and generally well explained.

The main problem with the paper is the English language which needs some fairly major changes.

I have detailed these below where possible to aid the authors. I have tried to suggest how to reword the bits which need change.

Line 19- please start the sentence with a capital letter

Line 19- demonstrated a high prevalence…. (reword)

Line 21- in vitro should be in italics

Line 21- development of biofilms … (reword)

Line 23- the bacterial name shortened is not required in brackets- this can be deleted

Line 24- ..in dairy animals … (reword)

Line 25- -administration of an antimicrobial agent… (reword)

Line 26- the ability of S. aureus …. (reword)

Line 31- to screen for subclinical … (reword)

Line 33- using a biofilm formation assay in a 12 well …. (reword)

Line 38- this is good, but can you make it clearer, maybe have the antibiotic name, and then (n = xx: xx%)

Line 41- weak biofilm producers …. (reword)

Line 41- interestingly is a typo

Line 41- the MDR isolates tended to form weak and moderate biofilms …. (reword)

Line 41- moreover is a typo

Line 42- delete ‘of’

Line 43- do you mean invisibility?

Line 44- isolated from subclinical mastitis … (reword)

Line 44- provides additional …. (Reword)

Line 50-51- wrong referencing style

Line 51- several microorganisms …(reword)

Line 53-54- the bacterial species need to be written in full on their first use

Line 55- Among them, S. aureus is the main bacteria responsible ….. (reword)

Line 56- delete ‘are’

Line 56- is the most common pathogen associated …. (Reword)

 Line 61- antimicrobial doesn’t need to be in italics

Line 61- treatment in the dairy industry …(Reword)

Line 65- this is repetitive, so please delete

Line 66- besides development of resistance to antimicrobials, S. aureus isolated from mastitis is also known to produce biofilms as well as being capable of invading and surviving within host cells’ …. (reword)

Line 71- the presence of S. aureus in cells provides the bacteria with a privileged, secure reservoir from which reinfection can occur (13), and this can result in long term chronic and reoccurring infections (14) …. (reword)

Line 81- tropical climate, characterised by high humidity, high rainfall ….. (reword)

Line 82 – wrong referencing style

Line 83-84- how randomly? Based on what information?

Line 87- swabbing of the teat end …. (reword)

Line 92- within 24 hours of arrival at the laboratory …(reword)

Line 100- a cut off of …. (reword)

Line 101 milk was from an animal having …. (reword)

Line 131- isolated form a high risk sources where antimicrobials are frequently used … (reword)

Line 134- Mac-T cells were used as the model of the bovine udder cells and the host for the intracellular infections in this experimental study (reword)

Line 140- in a cell culture flask … (reword)

Line 147 and 153- antimicrobials doesn’t need a capital

section 2.6 – what was the resistance profile of the bacteria used here?

Line 158- performed using S. aureus … (reword)

Line 162- 12 well cell culture plates …. (reword)

Line 177- somatic cell analysis …. (Reword)

Section 3.1. were any other bacteria tested for?

Line 182- showed a lower subclinical …. (reword)

Table 1- can you please add a space between the n number and the (30%). E.g. 17 (30%)

Line 188- the disc diffusion method …. (reword)

Line 188- to determine S. aureus ….(reword)

Line 189- towards the tested antimicrobials …. (reword)

Line 189- delete ‘the’ at the start of the sentence

Line 190  - this is good, but can you make it clearer, maybe have the antibiotic name, and then (n = xx: xx%)

Line 191- On the other hand…. (reword)

Line 192- space between aureus and were

Table 2- whats ZD? Please define the abbreviations

Line 198- 3 doesn’t need to be in capitals. And a full stop needed after 3

Line 199- MDR is usually resistant to 3 antibiotics?

Line 201- two is a typo

Line 203-205- This is unclear and I don’t know what you are trying to say so cant suggest a change

Line 209- please start sentence with a capital letter

Line 209- space between aureus and (n=30)

Line 210- maybe for the ease of description,…?

Line 212- please remove full stop before (Figure 2)

Line 21-2 And table 4 presents …. (reword)

Line 213- the biofilm formation ability of S. aureus …. (reword)

Line 215- respectively compared with 12.5% … (reword)

Line 223- formation and susceptibility are both typos

Line 226- Gentamycin is a typo

Line 227- were all different AMR profiles in here tested?

Line 231- why are these not visible in figure 2?

Line 235 and 236- please put S. aureus into italics

Line 236- space needed between F53D. The

Line 241- milk samples … (reword)

Line 244- please put in vitro into italics

Line 248- clinical symptoms displayed … (Reword)

Line 250- which is only usually  available in a reference laboratory and therefore restricts the ….. (reword)

Line 252- lateral flow tests have been developed for early …. (reword)

Line 264-266- This is unclear and I don’t know what you are trying to say so will struggle to offer an alternative wording. Please reword

Line 266- poor diets given to the animals. …. (reword)

Line269- this is repetitive. Please reword

Line 270- mastitis in animals…. (reword)

Line 273- one of the reasons for the failure. …(reword)

Line 278- this report is in agreement with previous…. (Reword)

Line 283- maybe reported may sound better than discovered?

Line 284- S. aureus to penicillin … (reword)

Line 285- were found to be invasive …(reword)

Line 291- S. aureus associated subclinical mastitis has been linked to the poor immune reaction …(reword)

Line 302- small molecule antibiotics and prevent it from reaching the individual cells ….(reword)

Line 307- delete ‘on’

Line 312- mastitis is a challenge in dairy …(reword)

Line 313-314- please reword this as it doesn’t make sense

Reviewer 2 Report

The introduction only repeats well-established knowledge. Hence, you should reduced in length.

Also, the hypothesis with regard of intracellular survival of S. aureus is totally out of context in this study. Please rewrite.

The MAR index is wrong. Please use a correct definition for multiple resistance. This index is based on older knowledge (early 1980’s) and should not be used anymore.

Please provide 95% CIs for all proportions.

In the discussion, please explain reasons for possible differences between provinces.

How did you decide on the breakpoints for antibiotic resistance-susceptibility?

Table 4 please move to supplementary material.

Please provide the biofilm production profile for all the isolates with multiple resistance.

Overall: this is a basic study that does not give any significant new knowledge – for acceptance, it needs a significant restructuring as above and also reduction in size and resubmission as communication.

Round 2

Reviewer 2 Report

The manuscript still remains of low quality and interest.

The authors make fundamental mistakes and do not show understanding of the multi-resistance to antimicrobial agents. This becomes evident by the use of the so-called MAR index.

A simple example of the errors of this index is the following.

Suppose that testing was performed to the following antibiotics
penicillin (R), ampicillin (R), amoxycillin (R), cefuroxime (R), streptomycin (S), enrofloxacin (S), gentamycin (S), erythromycin (S) à hence MR would be 4/8 = 0,5, whilst all antibiotics against which resistance was evident, were in the same class.

So, the authors show lack of basic knowledge of the topic and the manuscript can be misleading to future readers. Consequently, the manuscript is of low value.

Given the circumstances, I return an opinion of minor revision and I recommend rejection to the editor of the journal, who should be taking the final decision.
